# Biomarkers and Rehabilitation for Functional Neurological Disorder

**DOI:** 10.3390/jpm14090948

**Published:** 2024-09-07

**Authors:** Victor W. Mark

**Affiliations:** 1Department of Physical Medicine and Rehabilitation, Heersink School of Medicine, University of Alabama at Birmingham, Birmingham, AL 35294, USA; vwmark@uabmc.edu; Tel.: +1-205-934-3499; 2Department of Neurology, Heersink School of Medicine, University of Alabama at Birmingham, Birmingham, AL 35249, USA; 3Department of Psychology, College of Arts and Sciences, University of Alabama at Birmingham, Birmingham, AL 35294, USA

**Keywords:** functional neurological disorder, MRI, rehabilitation

## Abstract

Functional neurological disorder, or FND, is widely misunderstood, particularly when considering recent research indicating that the illness has numerous biological markers in addition to its psychiatric disorder associations. Nonetheless, the long-held view that FND is a mental illness without a biological basis, or even a contrived (malingered) illness, remains pervasive both in current medical care and general society. This is because FND involves intermittent disability that rapidly and involuntarily alternates with improved neurological control. This has in turn caused shaming, perceived low self-efficacy, and social isolation for the patients. Until now, biomarker reviews for FND tended not to examine the features that are shared with canonical neurological disorders. This review, in contrast, examines current research on FND biomarkers, and in particular their overlap with canonical neurological disorders, along with the encouraging outcomes for numerous physical rehabilitation trials for FND. These findings support the perspective endorsed here that FND is unquestionably a neurological disorder that is also associated with many biological markers that lie outside of the central nervous system. These results suggest that FND entails multiple biological abnormalities that are widely distributed in the body. General healthcare providers would benefit their care for their patients through their improved understanding of the illness and recourses for support and treatment that are provided in this review.

## 1. Introduction

Functional neurological disorder (FND) involves involuntary, intermittent neurological symptoms or signs that vary in relation to the patient’s self-attention to the symptoms or emotional excitation. Although “FND” is the term that was most recently professionally adopted for this illness, for centuries it was addressed by many other names (hysteria, conversion disorder, and psychogenic disorder among them [1]). The recent name change was recommended in 2014 by specialists who treat FND, to allay patients’ concerns for their being inappropriately diagnosed with a mental disease [2]. (“Functional neurological symptom disorder” is a widely used but much less common synonym).

This review is intended to examine whether FND can share biological characteristics with canonical neurological disorders. Where possible, comparison is made to specific canonical neurological disorders that share specific abnormalities that were uncovered in this article’s systematic literature review on FND. As a result, FND itself should be considered as a neurological disorder, as well that it is associated with multiple diverse biological abnormalities that extend beyond the central nervous system. In addition, although some reviews have concluded that FND has a poor prognosis, numerous recent neurorehabilitation trials for FND that were developed to treat canonical neurological disorders have shown favorable outcomes, which will be reviewed here. This review draws from systematic reviews of the published literature.

## 2. Clinical Characteristics of FND

Although FND has been noted to have a peak age of onset at about 40 years [3,4], it can begin anytime from childhood all the way to geriatric age [5,6,7,8,9]. Any voluntary activity can be affected by FND, including limb or facial movement, speech, cutaneous or muscular perception, and vision. The patients’ self-attention to their symptoms or emotional excitement can aggravate them, while distraction from them may reduce their severity [10]. FND symptoms can be provoked by direct medical examination and subside when the patient believes that he or she is not observed or undergoing formal evaluation [11,12,13]. The intermittent dysfunction is not apparently caused by epileptic brain discharges, transient cerebral ischemia, medication side effects, intoxication, metabolic or endocrinological diseases, systemic infection, or fatigue. As many as 18% of patients who are seen on neurological hospital wards can have FND [14].

FND has a high incidence of comorbid mood disorder, though this does not occur with all patients [9,15,16,17,18,19]. The disturbances are frequently considered to “mimic” canonical neurological disorders [20,21,22,23], which has often led practitioners to infer that the patients contrive their symptoms and thus that the disturbances do not constitute a neurological disorder [24]. FND lacks characteristic abnormalities on clinical structural brain or spinal imaging [25,26,27,28,29], although structural abnormalities that are not specific to other illnesses nonetheless can appear on clinical MRI [30,31]. Psychological care, particularly Cognitive Behavioral Therapy (CBT), can attenuate the symptoms [32]. For these reasons, FND has long been viewed by the public and clinicians as a mental illness in which the patients’ mood disorder is at the root of the disturbance [33], or that the symptoms are intentionally produced (malingering) for secondary gain [34]. In professional publications, FND is often regarded to be unable to be explained by commonplace neurology or general medicine concepts [35,36,37,38,39] and is widely considered to be distinct from “organic” disease [40,41,42]. Consequently, there is common stigmatization and social isolation for persons with FND [43,44].

However, starting about 30 years ago, the findings of distinct regional cerebral metabolic changes in physiological brain imaging studies in persons with FND, not found in neurologically healthy individuals [45,46], invigorated neuroscientific investigations in FND. These continue at an accelerating pace, as shown by the annual rate of publications for FND that are listed in the public registry of medical research publications, PubMed (https://pubmed.ncbi.nlm.gov, accessed 1 September 2024; Figure 1). Many of these studies suggest, instead, that FND has a biological basis. As will be shown here, many of these biomarkers occur also in canonical neurological disorders. Moreover, physical rehabilitative techniques that are commonly practiced for canonical neurological disorders (e.g., stroke, spinal cord injury, traumatic brain injury) have been shown similarly to benefit FND and are detailed later.

To support reviewing FND as a neurological disorder, this overview is organized into the following sections. Section 3. Biomarkers: The first subsection (Section 3.1) will review objective findings obtained from FND patients either during clinical evaluations while they were at rest or from tissue samples. The second subsection (Section 3.2) will review biomarkers from task-based neurophysiological evaluations. The third subsection (Section 3.3) will summarize behavioral findings in FND that are shared with canonical neurological disorders. Section 4 will review physical rehabilitation outcomes for motor FND, using methods that are extended from neurorehabilitation approaches that are widely used for canonical neurological disorders. These studies collectively suggest that FND is also a neurological disorder. This review will not address treatments for non-motor forms of FND (for example, functional seizures, functional sensory disorders, or functional cognitive disorders [38,47,48]), which rest mainly on psychological rather than physical treatments. These latter kinds of FND would require extensive additional discussion as to whether they may involve primarily psychological processes and would best be reserved for a separate report.

## 3. Biomarkers for FND

The systematic evaluation of published studies of biomarkers for FND in the present report was guided by previous FND biomarker literature reviews [1,49,50,51,52,53,54,55]. From these, the following Boolean search was run in PubMed: (functional neurological disorder OR functional neurological symptom disorder OR psychogenic disorder OR conversion disorder OR hysteria) AND (biomarker OR MRI OR positron OR single-photon OR diffusion tensor imaging OR DTI OR somatosensory evoked potentials OR genetics OR autonomic OR inflammation OR hypermobility syndrome OR accelerometry OR placebo OR endocrinologic disorder OR EMG OR electromyography). This yielded 3302 articles. These in turn were inspected for whether they were primary data reports of patients with FND, and excluded reviews, commentaries, and correspondence that referred to earlier articles. This step resulted in 102 included studies.

### 3.1. Objective Clinical Studies of Patients at Rest or from Tissue Samples

#### 3.1.1. Advanced Structural Brain Imaging Studies

Although distinctive findings do not occur in the individual clinical brain imaging study for persons with FND [56], statistical analyses of group-level data have distinguished persons with FND from neurologically healthy individuals. Structural brain MRI analysis, primarily using voxel-based morphometry, has generally indicated significant structural changes in brain gray areas. As of now, 21 studies have evaluated volumetric brain abnormalities in persons with FND (Table 1). The preponderance of the studies found reduced focal cortical or subcortical gray tissue areas when compared to individuals without FND. There was no pattern of volume loss that characterized FND.

For comparison, there has been no characteristic volume loss pattern in patients with Parkinson disease, a common neurodegenerative disease [78], even though numerous voxel-wise brain gray matter analyses consistently have shown cortical atrophy [79].

As shown by Table 1, alternative, less frequent patterns also occurred: (1) both volumetric focal decrease and increase in the same study group, (2) only focal volume increase, and finally, (3) no difference in brain regional volume compared to non-FND participants. Of note also is that for dystonia, for which there is evidence that it is a neurodegenerative disease [80], and Alzheimer disease, a leading neurodegenerative disease [81], quantitative brain morphological assessment has shown both focal decreased and increased volumes in the same populations [82,83]. Thus, strictly focal brain regional volume decrease is not characteristic of canonical neurodegenerative illness.

In addition, “histogram analysis” of the grayscale values in designated subcortical regions of interest on structural brain MRI has shown significant differences between persons with FND and neurologically healthy individuals [84,85]. This implies that significant histological characteristics reside in the basal ganglia of persons with FND, though the histological bases for these findings are thus far unknown.

Diffusion tensor imaging (DTI) is a complementary structural MRI assessment that evaluates the tendency for water molecules to diffuse either randomly or directionally constrained in neurological tissue. Net water diffusion in neurologically healthy individuals is less random compared to various neurological diseases [86]. Abnormal cerebral white matter DTI measures have been identified in many FND studies compared to healthy individuals [64,87,88,89], though this was not found in other studies [90,91,92]. The abnormalities generally involved reduced fractional anisotropy (representing more random diffusion) and increased mean or radial diffusivity values in select subcortical regions of interest. Similar abnormalities have been found in Parkinson disease, Alzheimer disease, essential tremor, orthostatic tremor, multiple sclerosis, and acquired (but not inherited) pediatric dystonia, among numerous other neurological disorders [93,94,95,96,97]. Caution with interpreting DTI studies is needed owing to the technique’s being sensitive to inadvertent head motion in the participants [98].

#### 3.1.2. Resting Brain Physiological Patterns

To identify central nervous system resting physiological patterns that would appear to distinguish FND from normal activity, the literature review found 12 such studies, which used functional MRI (fMRI), positron emission tomography (PET), single-photon emission computed tomography (SPECT), or somatosensory-evoked potentials.

Numerous brain imaging studies have identified in persons with FND the possibility to have abnormal resting regional physiology [45,46,99,100,101,102], somatosensory stimulation patterns [103,104,105], or intracerebral functional connectivity [106,107,108,109,110,111,112,113,114]. In many cases, regional hypometabolism had improved or been resolved in parallel with clinical recovery. Regional cerebral hypometabolism has also been identified in numerous canonical neurological disorders, including stroke, Alzheimer disease, parkinsonism, and corticobasal syndrome [115,116].

No somatosensory potentials could repeatedly be found at the scalps of two acute FND patients but which were found following full recovery at the 6-month follow-up [117].

#### 3.1.3. Genetic Analyses

Genetic bases for several neurodegenerative disorders have been identified, although these diseases more often are sporadic than familial. In Parkinson disease, 90 genetic risk factors have been identified [118]. Hereditability for multiple sclerosis has been well described [119]; the HLA DRB1*1501 haplotype has been most significantly associated with increased risk for the disease, among more than 200 other genes [120]. The finding of numerous genetic mutations in forms of dystonia in the later 20th century changed the neuroscientific view of dystonia from earlier decades, when it was thought to have a psychiatric etiology, to being a neurological disorder [121,122].

In contrast to this considerable body of study, research for specific genotypes in FND has markedly lagged. A tryptophan hydroxylase 2 gene polymorphism—*G703T*—has been shown to predict early-age onset of FND [123]. Functional seizures, also called paroxysmal or psychogenic nonepileptic seizures, are generally considered to be a kind of FND. Such seizures are associated with polymorphisms of the *FKBP5* gene, but only when co-occurring with depression [124].

#### 3.1.4. Low-Grade Inflammatory Biomarkers

Inflammatory biomarkers have been identified in diverse neurodegenerative disorders. Recurrent inflammation of the central nervous system is well known to be a major determinant of disability in multiple sclerosis [120]. Chronic inflammation in the central nervous system in this disease contributes to neurodegeneration through impairing remyelination [125]. Evidence of low-grade nervous system inflammation has been recently found in other neurodegenerative diseases, including Parkinson disease, Huntington disease, and amyotrophic lateral sclerosis, based on either measuring serum pro-inflammatory cytokines or identifying increased microglial activation in the brain on positron emission tomography [126,127].

Recent research has identified elevated serum cytokines as well in persons with FND, in particular, IL6, IL12, IL17A, IFNg, TNFa, and VEGF-a [128]. Elevated serum C-reactive protein levels have been identified in children and adolescents with FND [129]. Cerebrospinal fluid leukocytosis has been found in the majority of motor FND patients (*n* = 26) in a single study [130]. These findings thus far have not led to successful pharmacological trials of inflammatorily modifiable agents for FND.

#### 3.1.5. Non-Inflammatory Markers in Serum Samples

Brain-derived neurotrophic factor (BDNF), a growth factor, is fundamentally involved in neuronal recovery, neuroplastic reorganization, and brain development [131]. Reduced serum BDNF levels have been found in FND as well as in epileptic patients [132,133], which may be important for prognosis for clinical recovery. Low serum BDNF levels have also been found in numerous other canonical neurological disorders, including acute stroke, acute traumatic brain injury, Alzheimer disease, normal pressure hydrocephalus, Parkinson disease, and secondary progressive multiple sclerosis [134,135,136,137,138]. Use-dependent increase in BDNF levels, as can occur with physical rehabilitation [139], may be a potential biomarker for efficacious rehabilitation for FND.

#### 3.1.6. Autonomic Disturbance in Canonical Neurological Disease and FND

In some canonical neurological disorders, specific autonomic disturbances have been found that are shared with FND, which are indicated here.

##### Increased Resting Cardiac Contraction Rate

Tachycardia has been rarely characteristic in specific canonical neurological disorders. About 30% of patients with mitochondrial membrane-associated neurodegeneration have sustained tachycardia [140]. Orthostatic tachycardia has been found in patients with multiple system atrophy, another neurodegenerative disorder [141]. Some patients with functional movement disorder are often found also to have elevated heart rate at rest, including those with Postural Orthostatic Tachycardia Syndrome (POTS), compared to neurologically healthy control subjects [142,143,144,145].

##### Electrodermal Characteristics

Electrodermal activity can be influenced by eccrine gland releases in the skin, which are under autonomic nervous system control. When compared to epileptic patients, patients with functional seizures can demonstrate reduced electrodermal responses following an ictal event [146]. The findings preliminarily suggest that persons with functional seizures have less sympathetic arousal relative to persons with epilepsy following seizures.

#### 3.1.7. Clinical Electromyography

In suspected functional tremor, the diagnosis may be supported by the marked variability of the limb or axial muscle contraction frequencies at rest, by at least 1.5 Hz [147,148]. The specificity of this observation for FND, however, has not been evaluated.

#### 3.1.8. Gastrointestinal Motility Disturbances

Irritable bowel syndrome (IBS) refers to motility difficulties that can include irregular defecation and abdominal pain, without finding structural tissue abnormalities on standard imaging or scoping procedures after excluding inflammatory bowel disease [149]. Functional gastrointestinal disturbances may occur in as much as 41% of children [150] and 35% of adults with FND [151,152]. Similarly, various forms of functional (physiological) gastrointestinal motility impairments occur commonly in Parkinson disease, as much as 65% of patients, and the symptoms often long precede the onset of limb motor disturbances [153]. A similar proportion occurs in multiple sclerosis [154].

#### 3.1.9. Joint Hypermobility Disturbances

An unusually high prevalence (55%) of joint hypermobility has been reported in a sample of 20 FND patients [155].

The association between joint hypermobility and other neurological disorders has not been comprehensively examined, most likely in part because joint hypermobility is given little attention in formal medical training, and the finding is widely regarded as benign [156]. This latter view may overlook multiple organ dysfunctions that frequently accompany joint hypermobility, including gastrointestinal motility and cardiovascular autonomic disturbances.

Although general laxity of connective tissues could mechanically contribute to neurological disturbances due to compression of central nervous tissue, including from low-lying cerebellar tonsils in the type I Chiari malformation [157] and spinal instability [158], other neurological disturbances have no clear relationship to mechanical tissue disturbances. A case report of Ehlers–Danlos syndrome (a hypermobility disorder) identified co-existing limb myopathy on electromyography and ophthalmoplegia [159]. A sample of 90 individuals who scored abnormally high on a joint mobility assessment were found to have significantly reduced visual-evoked potentials latencies and amplitudes compared to individuals without excess hypermobility [160]. Joint hypermobility, therefore, may be a biomarker for extensive neurological dysfunction.

### 3.2. Task-Based Neurophysiological Studies

Because these studies require the patient’s careful following of instructions to perform specific tasks during brain physiological evaluation, they must be regarded with caution. Such studies have limited control over the patient’s understanding and compliance. In addition, repeated measures effects during physiological brain imaging can depend on the extent of patient stimulation [161,162], which can secondarily limit generalization of the conclusions from the studies.

PET or fMRI studies during specific tasks have identified significant differences between persons with FND and neurologically healthy control subjects, or between different tasks in the same FND individuals [163,164,165,166,167,168,169,170,171,172].

In patients who are considered to possibly have functional hemiparesis, transcranial magnetic stimulation can demonstrate reduced corticospinal excitability (changed latency and central motor conduction time in motor-evoked potentials) in the affected limb when compared to the unaffected limb, when patients are asked to imagine movement in the specific limb, as recorded by surface electromyography electrodes [173,174,175]. Consequently, this examination can demonstrate the differential effect of self-attention on central electrophysiological processes.

### 3.3. Behavioral Biomarkers of FND Shared with Canonical Neurodegenerative Disorders

#### 3.3.1. Clinical Blending between FND and Canonical Neurological Disease

Compelling research suggests that functional movement disorder can often evolve to canonical neurodegenerative disease. In a medical chart review, 26% of patients who were diagnosed with Parkinson disease (*n* = 53 total) had earlier developed FND [176]. An additional 8% of Parkinson disease patients had concurrent FND, and most of the Parkinson disease patients (57%) later developed FND. Similarly, Onofrj et al. as well as Pareés et al. in many instances observed FND to progress to either Parkinson disease or dementia with Lewy bodies [177,178,179]. Elsewhere, three cases of Creutzfeldt–Jakob disease were reported to have initially presented with functional movement disorder [180].

#### 3.3.2. Emotional Upset Effects on Symptoms

Emotional upset often provokes symptoms in FND [181]. Similarly, motor symptoms in Parkinson disease can be aggravated by anxiety or other emotional upset [182,183]. Anxiety and depression have been associated with worse performance on a standard test of visual information processing speed among persons with multiple sclerosis [184,185,186,187,188]. Fear of falling can aggravate postural control and gait control among persons with multiple sclerosis [189,190]. High anxiety is a risk factor for dystonic progression that starts with blepharospasm (involuntary contraction of eyelids) and then extends to other parts of the body [191].

#### 3.3.3. Exaggeration of Symptoms

Pareés et al. observed that persons with FND (*n* = 8) self-reported limb tremor that was more frequent than was captured by objective recordings of wrist-worn accelerometers worn in the home [192]. This finding suggests that persons with FND are prone to accentuated self-attention to their bodies, leading to their increased somatic complaints. This mismatch was greater than in a group of patients with “organic” tremor (not otherwise specified; *n* = 8) who wore the same instruments. Nonetheless, the latter group also exaggerated their amount of time with tremor when compared to the accelerometry data, thus showing that there is not an absolute difference between persons with FND compared to those with “organic” tremor with regard to symptom reporting.

A subsequent, slightly larger study by Kramer et al., using similar methods, found that while persons with FND reported more tremor disturbance than did persons with “organic” tremor, the self-reported “symptom burden” between persons with FND (*n* = 14) and those specifically with Parkinson disease (*n* = 6) did not differ [193]. The persons with “organic” tremor (including those with essential tremor, Parkinson disease, and other forms) were objectively recorded to have spent more time in tremor than those with FND, but the differences were slight. Though the findings did not support symptom exaggeration differences between persons with FND and those with other forms of tremor, it should be noted that the study was based on a small subject sample. These results, however, suggest that the differences between persons with FND vs. those with Parkinson disease are minimal with respect to subjective motor symptom impact.

#### 3.3.4. Expectation Effects on Symptoms

The research literature on FND extensively suggests that it is sensitive to suggestibility, i.e., placebo effects, though the studies appear to have had poor experimental design [194]. As far back as 1880, Charcot used hypnotic suggestion to modify symptoms in persons with FND [195]. Nonetheless, such suggestibility effects are by no means restricted to FND. Preliminary results suggest that hypnosis can improve motor symptoms in Parkinson disease or tics, though improved experimental controls were needed for the studies [196]. Placebo effects (expectation to improve) and the obverse, nocebo effects (expectation to worsen), are widely demonstrated in persons with Parkinson disease in reaction to treatments [197]. Low expectation can deter persons with Parkinson disease from pursuing physical exercise [198]. Similar effects are noted in many other involuntary movement disorders, including restless legs syndrome, Huntington disease, tics, amyotrophic lateral sclerosis, and multiple system atrophy [199,200,201].

#### 3.3.5. Context-Specific Changes on Locomotion

Patients with FND can improve their mobility during their formal rehabilitation by changing the method of locomotion. For example, an FND patient with impaired walking can improve by gliding the feet across the floor instead of lifting, as if moving across a slippery surface [202]. Limb movement ability in persons with FND can vary depending on whether muscle strength is formally tested vs. observing while the patient is walking [203]. Locomotion capability can vary depending on whether the patient walks across a level surface, compared to jogging, running, or using stairs [204]. Starting with a more stable form of locomotion, more complex activities can be gradually introduced as part of rehabilitation, which is described further below. Persons with Parkinson disease likewise can change their ability of locomotion by adopting different movement approaches or patterns. Walking backward or running can improve motor control in Parkinson disease, Huntington disease, or dystonia [205]. Freezing of gait in Parkinson disease can be improved by wearing shoes that project laser points of light in front of the wearer or by crawling on all four limbs [206,207]. Rhythmic auditory cues can improve voluntary movement in both Parkinson disease and FND [208,209]. Parkinson disease patients with freezing of gait may easily locomote by pedaling a bicycle on a street [210].

#### 3.3.6. Competing Voluntary Activities That Can Reduce Symptoms

As noted above under Section 2, Clinical Characteristics of FND, redirecting attention in persons with FND can ameliorate their symptoms [10]. Similar effects may occur with canonical neurological disorders. Classically, dystonia includes brief amelioration of the motor symptom through a self-initiated voluntary action by the patient, most often touching a specific part of the body. This behavior is commonly referred to as a “sensory trick” or “*geste antagoniste*” [211]. Although most often such *gestes* are simple, an inventory of such *gestes* shows that they can involve a wide variety of actions, including bending forward, yawning, wearing a scarf, cap, turban, or tight goggles, leaning on one’s elbows, picking at one’s teeth, singing, humming, drinking, kissing, whistling, chewing gum, laughing, piano playing, thinking about talking, running in a counterclockwise direction, listening to a loud radio, mirror viewing, or voluntary eye closure [212]. Such diversity raises consideration that these ameliorative actions may reflect the beneficial effect of redirecting attention from the predominant symptom, as suggested in the overviews of FND and Parkinson disease above.

#### 3.3.7. Cognitive Impairments

Cognitive impairments are common following canonical brain disease, as can be expected. These can also occur with FND. Among them are impaired memory [213], reduced processing speed [214,215,216,217], abnormal executive function [217,218,219,220,221], and impaired Theory of Mind (social cognition) [222,223]. In a single study, impaired executive function and Theory of Mind were shown to differ minimally between persons with FND and persons with Parkinson disease [223].

#### 3.3.8. Positive Response to Psychotherapy on Motor Symptoms

CBT, a form of psychotherapy, is a leading treatment for FND [224]. The treatment identifies events that trigger symptoms, diminishes attention to the impairment, redirects attention to better retained voluntary activities, cultivates self-efficacy, reduces emotional upset, and develops mindfulness (concentrating on current emotions and not focusing on events in the past or the future) [181]. The approach can reduce tremor severity and other motor symptoms in FND [225,226]. Although CBT is provided to persons with Parkinson disease mainly to control their mood disorders, preliminary findings indicate that the approach can also improve their walking [227]. Successful motor outcome also has been reported following CBT for cervical dystonia [228].

#### 3.3.9. Lower Extremity Dysesthesia and Compulsion to Move the Limbs

Restless legs syndrome involves annoying leg sensations (pain, tightness), most often while the patient is recumbent, and the compulsion to move the legs for relief. Increased leg movements also can occur in restless legs syndrome without leg discomfort, particularly during sleep. Restless legs syndrome considerably occurs in canonical neurological movement disorders, including Parkinson disease, multiple system atrophy, and multiple sclerosis [229,230].

Until the advent of advanced quantitative structural neuroimaging studies, restless legs syndrome was regarded primarily to be a “functional” (that is, physiological) disorder [231]. In a recent study of 96 individuals with functional movement disorder, the incidence of restless legs syndrome according to formal screening criteria was 44%, compared to 8% in neurologically healthy controls [232].

## 4. Physical Rehabilitation for FND

Until recent years, the long-term prognosis for FND was thought to be dismal [233,234,235]. Although numerous biomarkers for FND have been identified, these findings have not thus far indicated a consistently efficacious medical treatment for its impaired voluntary activities.

However, neurological rehabilitation has shown promise for controlling the symptoms of FND. In recent years, there has been increasing interest in developing and testing for neurological rehabilitation for FND, and transition from case series reports to larger clinical trials.

The greatest advances have been in applying neurological rehabilitation based on conventional methods toward controlling motor symptoms. Similar to the electronic literature review above, PubMed was searched with the terms (functional neurological disorder OR functional neurological symptom disorder OR hysteria OR psychogenic disorder OR conversion disorder) AND (physical therapy OR rehabilitation). The 891 entries were reviewed and excluded reviews, correspondence in response to other research, and studies that included five or fewer participants. Table 2 and Table 3 summarize the resulting trials (*n* = 35) in chronological order up to the present that applied to functional movement disorders. This summary encompasses more than 1500 individuals (mostly adults, but also children) who were treated and followed for the durations of the trials.

The published studies generally had favorable outcomes and, in many cases, gains retained over months or years. A considerable limitation among the studies has been the common lack of comparing one treatment to another in groups that were matched for the degree of disability. In addition, in most reports, patient groups had diverse symptoms that were targeted for treatment, leaving unclear whether treatment outcomes depended on the particular symptoms being treated.

Although the approaches somewhat differed from each other, a common approach was to start by having patients practice voluntary movements that entail little difficulty and can be accomplished successfully, and then advance gradually through more complicated movements, with praise at every stage of accomplishment [271]. This is subsumed under the term “shaping” [243,263,272,273,274], which has also been used in specific forms of physical rehabilitation (e.g., Constraint-Induced Movement Therapy) for canonical neurological disorders, including stroke, cerebral palsy, traumatic brain injury, and multiple sclerosis [275,276,277,278]. In addition, because FND symptoms are affected by self-attention to the deficits, the rehabilitation techniques emphasized increasing general physical activity without drawing attention to the particular part of the body or context, which could aggravate the symptoms. An example for the effect of attention on an FND deficit is the Hoover sign [279]. This involves the inability to extend a hemiparetic leg following direct command while supine or seated, but can occur when the patient is asked to redirect self-attention to the opposite leg and elevate it. In that case, the affected leg’s extension is necessary to stabilize the pelvis during the maneuver. Such a demonstration of retained movement capability when self-attention is redirected can serve as a basis for rehabilitation.

## 5. Other Treatments

To a lesser extent, other treatments for FND have been investigated. Because these methods are early in their development, the treatment results are not provided here. The methods have included CBT alone [225], hypnosis [280], Faradic stimulation to limb muscles [281], and transcranial magnetic stimulation [282]. Although various psychotropic medications have been tried for FND and can help to manage mood disorder, there is thus far no leading efficacious pharmacological treatment for the FND symptoms themselves [283,284].

## 6. Discussion

Reviews of FND biomarkers until now have focused either on motor or ictal forms of FND or subsets of evidence (neuroimaging, serum assays, behavioral measures) [1,49,50,51,52,53,54,55,285,286]. In contrast, the present review examines the biomarker evidence across diverse forms of FND and from a wider array of assessments. Moreover, this review is distinguished by its comparison to diverse canonical neurological disorders that share numerous clinical and laboratory-based findings. This review amply demonstrates that FND, a much misunderstood illness, shares many objective laboratory and clinical characteristics with canonical neurological diseases.

Limitations of this review are that a single reviewer conducted the literature search, which was based only on PubMed. In general, comprehensive literature reviews currently enlist multiple reviewers who compare their searches mutually and reach consensus for which articles should be included and the conclusions drawn. Commonly, multiple medical literature databases are searched in addition to PubMed (e.g., Embase, Web of Science, Scopus). Nonetheless, the present search method led to identifying multiple categories of FND biomarkers after consulting the several previously published FND biomarker reviews. The resulting categories were then checked to determine whether any of those could be shared with canonical neurological disorders; many were found. Consequently, this limited search method succeeded in identifying substantial overlap between FND and canonical neurological disorders. It is unlikely that a more extensive literature search method would have substantially changed the outcomes.

The results suggest that FND is a neurological disorder, in view of its morphological abnormalities demonstrated in numerous brain imaging studies and considerable behavioral characteristics that are shared with canonical neurological disorders, including the many instances of clinical transition between FND and other neurological disorders. Moreover, the results indicated many instances in which acute focal cerebral hypometabolism in persons with FND receded in parallel with clinical improvement. These sources of evidence imply that FND is a neurological disorder.

In addition, this review identifies FND biomarkers that involve many biological systems outside of the central nervous system, including cardiovascular, gastrointestinal, autonomic, immunological, and orthopedic systems, along with distinct genotypes that predict forms of FND. Thus, FND presents a complex medical illness that is associated with extensive abnormalities in the body. In this diverse presentation of FND, similarities can occur in other neurological disorders. One example is Parkinson disease, which has been shown to have characteristic findings in immunological, gastrointestinal, and genetic systems, as indicated above.

Furthermore, conventional physical neurorehabilitation techniques, which are used for chronically disabling canonical neurological disorders, can also ameliorate FND symptoms. This review thus may help to demystify the illness and encourage clinical practitioners to approach FND patients empathetically and supportively. This evidentiary foundation allows practitioners to indicate to their patients that (1) the illness is not fundamentally a mental disorder, and (2) the illness can respond positively to rehabilitation techniques that are similarly applied to other neurological disorders. In doing so, this review aims to assist FND into mainstream neurological care, to regard it as a neurological disease, and not to treat it as a fringe, exotic, or mysterious illness.

Even more importantly, the many biomarkers that FND shares with multiple other neurological disorders should prompt clinicians who evaluate and treat FND to be aware of and routinely evaluate for its multiple organ comorbidities. Of note, the constellation of autonomic, cardiovascular, immunological, gastroenterological, and orthopedic disturbances are not unique to FND. In recent research, this pattern also was found to be common in patients who presented with gastrointestinal motility disturbances that lacked observable tissue pathology, who were not considered to have FND but who were nonetheless found to have immunological, autonomic, and orthopedic abnormalities [287]. Consequently, treating an FND patient warrants investigating these possible other disturbances and consulting specialists in these areas where needed. The diverse biomarkers suggest that FND may not strictly be a neurological disorder. Although at present there are no clear physiological or developmental biological processes that may underlie FND, these findings may encourage further hypothesis development and clinical investigation to clarify the pathological processes that are involved with FND.

Evaluating and directing treatment for FND requires the expertise of a neurologist, owing to the complexity of the symptoms [288,289,290,291,292]. The optimal management of FND would start with accurate diagnosis. However, there are many difficulties with doing so:

(1) There is no gold standard for diagnosing FND. In our review of studies of FND biomarkers [51], we found that there are three main rival methods: the Fahn–Williams method and its variants [269], the method outlined in the various editions of the Diagnostic and Statistical Manual of Mental Disorders [293], and the referring physician’s personal judgment. Without a consensus diagnostic method for FND, rapid progress in research for treatment will likely be hindered.

(2) Seldom considered has been the extensive list of alternate paroxysmal neurological disorders that are not known to be affected by self-attention or emotional excitation, and which lack distinctive features on conventional clinical neuroimaging. These include frontal lobe epilepsy, paroxysmal dyskinesia, and autoimmune encephalitis [294]. This list obliges the involvement of a neurologist who is highly experienced with assessing FND vs. the alternate neurological disorders, thus, to guide the patient to appropriate management.

(3) Even when a neurologist with expertise in FND may be involved, current clinical practice often limits the time to evaluate new patients to 30 min, due to economic pressures and meeting the demands of a large practice [294]. In contrast, as much as an hour is necessary to conduct a thorough historical intake and comprehensive neurological evaluation, and to provide empathetic patient and family education and care planning. Furthermore, the patient’s concurrent cognitive limitations (described in Section 3.3.7) can limit or slow these steps. As a result, the modern medical practice milieu may prolong evaluating and ultimately arranging care for persons who may have FND or other paroxysmal disorders.

(4) Appropriate care, including treatment by a neuropsychologist and rehabilitative physical therapists, occupational therapists, or speech-language pathologists, can be limited because of the relatively few medical centers that can provide these services with commensurate expertise with FND. Consequently, there can be a considerable waitlist for patients to be seen, along with the hardship involved to arrange travel to such places.

To assist the management by the clinician who initially sees persons who may have FND, the web site FND Hope–FND Hope International (https://fndhope.org, accessed on 1 September 2024) lists such centers that have the available expertise. In addition, referring the patient and family to the web site https://neurosymptoms.org/en/ (accessed on 1 September 2024) can acquaint them with the diverse appearances and complexities of FND. Doing so can equip them with the knowledge to better understand the illness, which in turn could help to reduce the time for evaluation and allay concerns by confirming that the illness is not a mental disorder or a different neurological disorder with episodic symptom aggravation, such as multiple sclerosis [295,296].

## 7. Conclusions

This overview of the biomarkers and favorable responses to physical neurorehabilitation for FND implies that it is a neurological disorder. Consequently, the patient should be approached with this in mind, following appropriate diagnosis. Becoming familiar with the content of this review can prepare the clinician to approach the patient with confidence that FND is not an unknowable, enigmatic disorder. Optimism should be conveyed to guide patients toward improved self-control of their symptoms with competent rehabilitation.

## Figures and Tables

**Figure 1 jpm-14-00948-f001:**
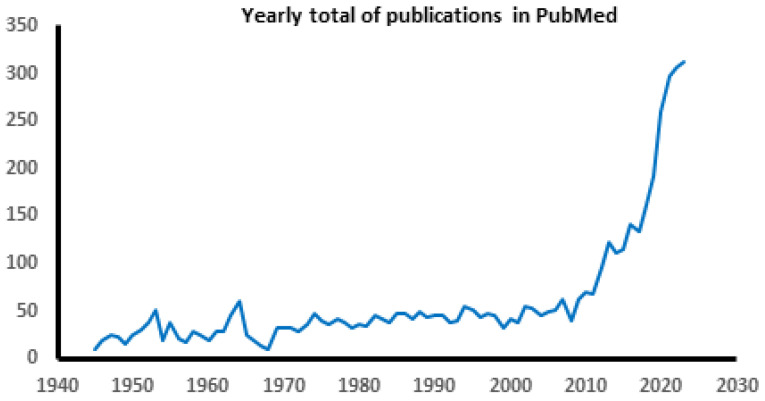
Yearly total of publications appearing in PubMed that included the terms “functional neurological disorder”, “functional neurological symptom disorder”, or “conversion disorder”. The alternate terms “hysteria” and “psychogenic disorder” are not included in this graph because they include other disorders that are not FND in addition to FND.

**Table 1 jpm-14-00948-t001:** Summary of volumetric gray area changes on structural brain imaging in FND.

Volume Findings	Number of Studies	References
Focal decrease	11	[57,58,59,60,61,62,63,64,65,66,67]
Focal decrease and increase in different areas	4	[68,69,70,71]
Focal increase	3	[72,73,74]
No difference compared to non-FND subjects	3	[75,76,77]

**Table 2 jpm-14-00948-t002:** Summary of physical therapy for FND: diagnostic methods, interventions, doses, settings, and targeted symptoms.

Study [Reference]	*n* *	Diagnostic Method **	Intervention †	Dose (Months)	Setting ††	Motor Symptoms Targeted ‡
Weiser, 1976 [236]	7	MD referral	PT, counseling	0.25–2	Out	paresis
Delargy, 1988 [237]	6	MD referral	PT	0.36–2.5	In	walking
Leslie, 1988 [238]	20	MD referral	PT	1–3	In or Out	walking, paresis
Speed, 1996 [239]	10	MD referral	PT	0.14–0.8	In	walking
Heruti, 2002 [26]	30	MD referral	PT	Not stated	In	paresis
Moene, 2002 [240]	45	DSM-III	PT + hypnosis vs. PT	3	In	dystonia, walking, tremor, paresis
Schrag, 2004 [235]	26	Fahn–Williams	PT + CBT	Not stated	Not stated	dystonia, tremor
Schwingenschuh, 2008 [241]	12	Fahn–Williams	PT + CBT	Not stated	Out	dystonia, walking, tremor
Dallochio, 2010 [242]	16	Fahn–Williams	Walking therapy	3	Out	dystonia, walking, tremor
Czarnecki, 2012 [243]	80	Fahn–Williams	PT vs. TAU	0.25	Out	walking, tremor, paresis
Saifee, 2012 [244]	26	MD referral	PT + CBT	0.75	In	dystonia, tremor, paresis
Kozlowska 2013 [245]	56	MD referral	Multidisciplinary rehab	1	In and Out	NA
Demartini, 2014 [246]	36	MD referral	Multidisciplinary rehab	1	In	dystonia, walking, tremor, paresis
Espay, 2014 [247]	10	MD referral	Entrainment with biofeedback device	0.03	Out	tremor
Jordbru, 2014 [248]	40	MD referral	PT + CBT vs. waitlist	0.75	In	walking
McCormack, 2014 [249]	33	MD referral	Multidisciplinary rehab	3.3	In	dystonia, tremor, paresis
Nielsen, 2015 [250]	47	Gupta–Lang	PT	0.25	Out	dystonia, walking, tremor, paresis
Dallochio, 2016 [251]	29	Fahn–Williams	CBT vs. CBT + PT vs. TAU	3	Out	dystonia, walking, tremor
Matthews, 2016 [252]	35	MD referral	PT	≤2	In	walking
Nielsen, 2017 [253]	57	Fahn–Williams	PT vs. nonspecific PT	0.25	Out	dystonia, walking, tremor
Jacob, 2018 [254]	32	Fahn–Williams	PT	0.25	In	dystonia, walking, tremor
Jimenez, 2019 [255]	49	DSM-5	Pain multidisciplinary rehab	0.25	Out	FMD
Bullock, 2020 [256]	12	DSM-5	VR motor rehab + mirror feedback	2	Out	FMD or sensory symptoms
Demartini, 2020 [257]	18	Gupta–Lang	PT	5.25	Home	dystonia, walking, tremor, paresis
Maggio, 2020 [258]	50	DSM-5	PT, CBT, goal setting	1.5–3	Out	dystonia, walking, tremor, paresis
Petrochilos, 2020 [259]	78	MD referral	PT, multidisciplinary rehab, CBT	1.4	Out	dystonia, walking, tremor, paresis
Gandolfi 2021 [260]	33	Gupta–Lang	PT	0.25	Home	dystonia, walking, tremor, paresis
Reid 2022 [261]	18	MD referral	Multidisciplinary rehab	0.25	Out	not specified
Hebert, 2021 [262]	13	Fahn–Williams	PT	0.25–0.5	In	dystonia, walking, tremor, paresis
Callister, 2023 [263]	201	Gupta–Lang	PT	0.25	In	walking, tremor, paresis
Guy, 2024 [264]	31	DSM-5	PT + CBT	2	Out	dystonia, tremor, paresis
McCombs, 2024 [265]	77	MD referral	Sensory-oriented OT	4	Out	dystonia, walking, tremor, paresis
Nielsen, 2024 [266]	241	Gupta–Lang	PT vs. TAU	0.75	Out	dystonia, walking, tremor, paresis
Polich, 2024 [267]	22	MD referral	PT	0.5	In	walking, paresis
Macías-García, in press [268]	38	Gupta–Lang	PT + CBT vs. psychol support	1.5	Out	dystonia, walking, tremor

* *n*, number of patients who completed the study. ** Diagnostic method: DSM, Diagnostic and Statistical Manual of Mental Disorders (various editions); Fahn–Williams [269]; Gupta–Lang [270]; MD referral, clinician referral. † Intervention: CBT, Cognitive Behavioral Therapy; OT, occupational therapy; PT, physical therapy; TAU, treatment as usual; VR, virtual reality. †† Setting: In, inpatient; Out, outpatient; Home, home-based therapy. ‡ Motor symptoms targeted: FMD, functional movement disorder not otherwise specified. NA, not reported.

**Table 3 jpm-14-00948-t003:** Summary of physical therapy for FND: outcome measures, immediate results, follow-up, and follow-up results.

Study	Outcome Measure ‡‡	Immediate Results	Follow-Up (Months) ¶	Results ¶¶
Weiser, 1976 [236]	Neurol exam	100% improved	1–96	86% retained gains
Delargy, 1988 [237]	Neurol exam	100% improved	8–14	100% retained gains
Leslie, 1988 [238]	Neurol exam	85% improved	NA	
Speed, 1996 [239]	FIM	100% improved	7–36	78% retained gains
Heruti, 2002 [26]	Neurol exam	55% improved	NA	
Moene, 2002 [240]	Neurol exam	65% improved; no difference between groups	6	84% retained gains; no difference between groups
Schrag, 2004 [235]	Neurol exam	33% improved	NA	
Schwingenschuh, 2008 [241]	Neurol exam	80% improved	NA	
Dallochio, 2010 [242]	PMDRS	70% improved	NA	
Czarnecki, 2012 [243]	Neurol exam	73% improved	25–33	Experimental group 60% self-rated improved vs. 22% control treatment
Saifee, 2012 [244]	Nonspecific self-assessment	58% improved	NA	
Kozlowska 2013 [245]	Neurol exam	63% improved	NA	
Demartini, 2014 [246]	COPM, CGI	67% improved	12	COPM: 100% retained gains; CGI: 33% retained gains
Espay, 2014 [247]	PMDRS	100% improved	3–6	50% retained gains; the other measures declined
Jordbru, 2014 [248]	Functional Mobility Scale, FIM	Experimental group generally improved	12	100% gains retained
McCormack, 2014 [249]	Mobility, MRS	Generally improved	NA	
Nielsen, 2015 [250]	CGI	96% improved	3	85% retained gains
Dallochio, 2016 [251]	PMDRS	Experimental groups improved, unlike TAU	NA	
Matthews, 2016 [252]	Modified Rivermead Mobility Index	Generally improved	NA	
Nielsen, 2017 [253]	CGI	Assessment delayed until 6 m	6	Experimental group > control group gains
Jacob, 2018 [254]	CGI, PMDRS	87% improved	6	67% retained (only CGI assessed)
Jimenez, 2019 [255]	In-lab movement measures	Generally improved	NA	
Bullock, 2020 [256]	Oxford Handicap Scale	Improved experimental group only	NA	
Demartini, 2020 [257]	PMDRS, CGI	Improved (67%)	6	72% retained gains
Maggio, 2020 [258]	Subjective change	Improved (34%)	NA	
Petrochilos, 2020 [259]	CGI, COPM	Generally improved	6	100% retained gains
Gandolfi 2021 [260]	S-FMDRS, other in-lab measures	Generally improved	3	Gains lost
Hebert, 2021 [262]	CGI	93% improved	12	77% retained gains on CGI
Reid 2022 [261]	COPM, lab assessments	Generally improved	NA	
Callister, 2023 [263]	COPM ability subscale	84% improved	NA	
Guy, 2024 [264]	lab assessments	Generally improved	3	100% retained gains
McCombs, 2024 [265]	clinician judgment	62% improved	NA	
Nielsen, 2024 [266]	SF-36; CGI	Not reported	12	No difference between groups on SF-36; results favored CGI, but statistics not stated
Polich, 2024 [267]	PT, OT judgment (ordinal scales), Berg Balance Scale	95% improved	NA	
Macías-García, in press [268]	SF-36; CGI; EQ-5D	Greater improvement in experimental vs. control	5	Partial regression

‡‡ Outcome measure: CGI, Clinical Global Impression self-rated scale; COPM, Canadian Occupational Performance Measure; EQ-5D, quality of life at 5 months post-treatment; FIM, Functional Independence Measure; Mobility, nonspecific assessment of walking; MRS, Modified Rankin Scale; Neurol exam, neurological examination; PMDRS, Psychogenic Movement Disorder Rating Scale; S-FMDRS, Simplified Functional Movement Disorder Rating Scale. ¶ Follow-up: NA, not reported. ¶¶ Results: SF-36, 36-item Short Form Health Survey.

## Data Availability

No new data were created or analyzed in this study.

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
