# Peer review of "Biomarkers and Rehabilitation for Functional Neurological Disorder"

_jpm, 2024, doi:10.3390/jpm14090948_

Round 1

Reviewer 1 Report

Comments and Suggestions for Authors

This review is based partly on the author’s experience as director of a dedicated FND Clinic, partly on a reading of the literature.

The argument sometimes advanced that symptoms which cannot be explained on a neurological basis, and therefore termed functional, or psychogenic, or non-organic, are hence not biological or physiological in origin, is of course merely a cloak for clinician ignorance.  [The terminology is problematic, the author himself equating “functional” with “physiological” (Line 388).]  Many disorders once considered functional are now reclassified as neurological (e.g. dystonia, Tourette’s).  The author is therefore right to draw attention to possible biomarkers of FND which might give insights into the biology, or pathophysiology, or psychopathophysiology, of these highly prevalent conditions. 

However, the listing of symptoms and investigation findings in certain neurological disorders and which may also be found in FND does not really help, e.g. the long description of MPAN (Lines 146-152) as possible justification for a biological cause of “increased cardiac contraction rate” in FND.  Surely it would be more profitable to investigate the causes of “increased cardiac contraction rate” in FND patients, if such exists?  The section on structural brain imaging (Lines 91-111) does attempt to address FND in this way, although not in adequate depth.  A similar approach to functional brain (magnetic resonance) imaging would be of particular interest.

The author’s willingness to equate the current category of “FND” with previous nosological categories, such as hysteria or conversion disorder, is historically questionable.  Even if there were some clinical overlap, it would still be untenable to credit Freud or Charcot or Kinnier Wilson retrospectively with insights into “FND” as now characterised.

Although designated a “review” paper, no review methodology, systematic or narrative, is stated.  The methodology needs to be clarified.  Much of the evidence presented appears to be at the lowest (“anecdotal”) level.  Perhaps the paper might better be classified as a “perspective” piece?

Specific points:

Lines 57-58: “which may lead to falsely infer that the patients contrive their symptoms.” As written, this is not grammatically correct.  Perhaps to read “which may lead to the incorrect inference that patients contrive their symptoms”.

Lines 60-61: “The “conversion disorder” hypothesis for FND was postulated by the noted Austrian psychiatrist Sigmund Freud (1856-1939)”. This is not tenable, since anachronistic: Freud would not have known what “FND” was. Certainly symptoms were labelled as “functional mental illness” or “functional mental disorder” during his era, but it is historically naïve to assume that these categories may be simplistically equated.

Line 70: “FND (then termed hysteria)”. See previous comment.

Line 93: “brain grey areas” to read “brain grey matter areas”.

Line 175: “the author has observed the phenomenon to be even more frequent”. Data would be preferable to bald assertion.

Lines 205-206: “Samuel Alexander Kinnier Wilson (1878-1937), who rejected the psychiatric basis for FND”. See comments with respect to Freud, Charcot.

Lines 325-326: “Charcot applied hypnotic suggestion to modify symptoms in persons

with FND”. Anachronism.

Lines 456-457: “to discard the unsupported implication that it can corrupt reality testing”. Meaning unclear to me: what is “reality testing”?

Line 459: “aims to bring FND into mainstream neurological care”. Already the case, surely? Certainly in my experience of neurology in the UK.

Lines 468-469: “and consult specialists in these areas as needed.”. Is this wise? To risk further self-attention to symptoms?

Lines 514-515: “This overview of the biomarkers and responses to conventional neurological rehabilitation for FND demonstrates that it is a neurological disorder.” cf. Lines 469-470 “The diverse biomarkers also suggest that FND may not strictly be a neurological disorder.”. Which demonstrates that trying to pigeonhole the symptom complex to a specific clinical discipline doesn’t really work. The overlap of symptoms between organ systems (Lines 463-464: “autonomic, cardiovascular, immunological, gastroenterological, and orthopedic disturbances”) surely also supports this?

Comments on the Quality of English Language

No comments

Author Response

This review is based partly on the author’s experience as director of a dedicated FND Clinic, partly on a reading of the literature.

 The argument sometimes advanced that symptoms which cannot be explained on a neurological basis, and therefore termed functional, or psychogenic, or non-organic, are hence not biological or physiological in origin, is of course merely a cloak for clinician ignorance.  [The terminology is problematic, the author himself equating “functional” with “physiological” (Line 388).]  Many disorders once considered functional are now reclassified as neurological (e.g. dystonia, Tourette’s).  The author is therefore right to draw attention to possible biomarkers of FND which might give insights into the biology, or pathophysiology, or psychopathophysiology, of these highly prevalent conditions. 

 However, the listing of symptoms and investigation findings in certain neurological disorders and which may also be found in FND does not really help, e.g. the long description of MPAN (Lines 146-152) as possible justification for a biological cause of “increased cardiac contraction rate” in FND.  Surely it would be more profitable to investigate the causes of “increased cardiac contraction rate” in FND patients, if such exists?  

The description of MPAN is now reduced (lines 212-213). The point made here is that a few canonical neurological disorders have an elevated resting heart rate; similar is found in many FND patients. Further investigation will be needed to tease apart the source for tachycardia in these various populations.

The scientific study of FND did not take off until the past 20 years, as demonstrated by Figure 1. Causes of the various physiological and structural characteristics of FND have only minimally been investigated, and surely deserve more attention, one hopes.

The section on structural brain imaging (Lines 91-111) does attempt to address FND in this way, although not in adequate depth.  A similar approach to functional brain (magnetic resonance) imaging would be of particular interest.

 The sections on neuroimaging are now expanded to include physiological (or “functional”) brain imaging studies. The concern for devoting adequate depth is balanced with the total of the length of the review, which is being managed and not attempting to overwhelm one section with another.

The author’s willingness to equate the current category of “FND” with previous nosological categories, such as hysteria or conversion disorder, is historically questionable.  Even if there were some clinical overlap, it would still be untenable to credit Freud or Charcot or Kinnier Wilson retrospectively with insights into “FND” as now characterised.

 In response, I have taken out the historical line as to the development of the concept of FND. Note however that numerous reviews link hysteria, conversion disorder, psychogenic disorder, and ultimately FND. For example, Baumans J, Scantamburlo G. Des idées reçues sur l’hystérie au trouble neurologique fonctionnel. Rev Med Lìege 2023;78:261-266; Callister MN, Stonnington CB, Cuc A, et al. In patients with functional movement disorders, is specialized physical therapy effective in improving motor symptoms?: a critically appraised topic. Neurologist 2022;27:82-88; Broussolle E, et al. History of physical and 'moral' treatment of hysteria. Front Neurol Neurosci 2014;35:181-197.

The reference to Kinnier Wilson is now also removed, because it was a single observation that has not been confirmed.

Although designated a “review” paper, no review methodology, systematic or narrative, is stated.  The methodology needs to be clarified.  Much of the evidence presented appears to be at the lowest (“anecdotal”) level.  Perhaps the paper might better be classified as a “perspective” piece?

The point is well made. The author now applies a systematic literature review, using Boolean search methods in PubMed. I believe that the review now is strengthened, and I thank the astute reviewer.

 Specific points:

Lines 57-58: “which may lead to falsely infer that the patients contrive their symptoms.” As written, this is not grammatically correct.  Perhaps to read “which may lead to the incorrect inference that patients contrive their symptoms”.

 The recommendation is better than the original passage, and so will be used instead.

Lines 60-61: “The “conversion disorder” hypothesis for FND was postulated by the noted Austrian psychiatrist Sigmund Freud (1856-1939)”. This is not tenable, since anachronistic: Freud would not have known what “FND” was. Certainly symptoms were labelled as “functional mental illness” or “functional mental disorder” during his era, but it is historically naïve to assume that these categories may be simplistically equated.

What exactly Freud considered to be FND as it is now considered in modern terms, compared to a century or more ago, cannot be ascertained. To avoid the inability to ascertain the equivalence among hysteria, conversion disorder, etc, the historical development section is removed. Refer to the citations provided above for further background.

Line 70: “FND (then termed hysteria)”. See previous comment.

This line is now removed.

Line 93: “brain grey areas” to read “brain grey matter areas”.

The phrase is corrected now, thank you, as provided as the title of Table 1.

Line 175: “the author has observed the phenomenon to be even more frequent”. Data would be preferable to bald assertion.

This is an ongoing study, and so it would be inappropriate for me to provide an updated statistic. This statement is now removed.

Lines 205-206: “Samuel Alexander Kinnier Wilson (1878-1937), who rejected the psychiatric basis for FND”. See comments with respect to Freud, Charcot.

As noted above, the citation about Kinnier Wilson is now removed.

Lines 325-326: “Charcot applied hypnotic suggestion to modify symptoms in persons

with FND”. Anachronism.

The word FND is now replaced with the word hysteria for historical accuracy (lines 423-424). Note, as above, that many references link the historical line of terms hysteria, then conversion disorder, then psychogenic disorder, and now FND.

Lines 456-457: “to discard the unsupported implication that it can corrupt reality testing”. Meaning unclear to me: what is “reality testing”?

That vague phrase is removed now.

Line 459 (now line 565): “aims to bring FND into mainstream neurological care”. Already the case, surely? Certainly in my experience of neurology in the UK.

One should envy the sophistication of the UK medical care system. In the US, particularly in Alabama, patients diagnosed with FND are bounced around among medical practitioners who are given the summary statement, “It’s all in your head,” and even worse. Indeed, in a Southern medical school hospital, where this author practices, still many physicians infer without question that the FND patients had been sexually abused young in life, but which they have forgotten or repressed. This contention is not supported. Cf Burke MJ. "It's all in your head"—medicine's silent epidemic. JAMA Neurol 2019;76:1417-8.

Lines 468-469 (now line 578): “and consult specialists in these areas as needed.”Is this wise? To risk further self-attention to symptoms?

Yes, it is wise, ethical, and being medically responsible for a practitioner. As noted in the present manuscript, diagnosing and managing FND is best handled by a neurologist with expertise with this disorder. The approach to managing such patients requires considerable empathy and explanation, as for example demonstrated in Carson A et al. Explaining functional disorders in the neurology clinic: a photo story. Pract Neurol 2016;16:56-61 and Adams C et al. You've made the diagnosis of functional neurological disorder: now what? Pract Neurol 2018;18:323-30. The FND specialist can and should explain the disorder to patients in relation to the latest research that implicates it as (in part) a neurological disorder. Such FND specialist can demonstrate to the patient how self-attention can aggravate the symptoms, while distraction can ameliorate the symptoms, thus empowering the patient with a start for management. The present review draws attention to the multi-organ complications associated with FND, and thus to interpret for the patient how the particular extra-neurological disturbances can be treated. Many of the author’s patients bring to him these various multiple-organ disturbances, and so the patient already is attentive to and aware of these. Routinely, this author refers patients to orthopedic surgeons, rheumatologists, cardiologists, and gastroenterologists as needed. To ignore these multiple-organ manifestations would be irresponsible by a caring physician.

Lines 514-515: “This overview of the biomarkers and responses to conventional neurological rehabilitation for FND demonstrates that it is a neurological disorder.” cf. Lines 469-470 “The diverse biomarkers also suggest that FND may not strictly be a neurological disorder.” Which demonstrates that trying to pigeonhole the symptom complex to a specific clinical discipline doesn’t really work. The overlap of symptoms between organ systems (Lines 463-464: “autonomic, cardiovascular, immunological, gastroenterological, and orthopedic disturbances”) surely also supports this?

Pigeon-holing has its obvious limitations. One can peruse neurological texts to find the many dermatological manifestations of what is considered to be neurological disease, as for example of many neurocutaneous disorders (eg, Hansen’s disease, tuberous sclerosis, Sturge-Weber syndrome, various phakomatoses; cf Kioutchoukova I et al. Neurocutaneous diseases: diagnosis, management, and treatment [review]. J Clin Med 2024;13:1648), or hepatocerebral or hepatolenticular disorders (eg, Wilson disease, acquired hepatocerebral degeneration), or CNS disturbances associated with rheumatological disorders (cf Smiyan S, et al. Central nervous system manifestations in rheumatic diseases. Rheumatol Int, in press.) Thus, one may find such topics in neurological disorder texts, or alternatively in dermatology, hepatology, or rheumatological texts.

Categorizing disorders under a specific organ system is a convenient way to start medical care, but doing so can indeed misrepresent such multi-organ disorders as being strictly neurological.

I thank the reviewer for pointing out the self-contradiction in the lines indicated above here. In response, the sentence “This overview of the biomarkers and responses to conventional neurological rehabilitation for FND demonstrates that it is a neurological disorder” as the leading sentence for the section Conclusion (line 625) is now changed to read, “This overview of the biomarkers and responses to physical neurorehabilitation for FND suggests in part that it is a neurological disorder.”

Reviewer 2 Report

Comments and Suggestions for Authors

This manuscript, entitled “Biomarkers and Rehabilitation for Functional Neurological Disorder,” deals with a review of functional neurological disorders. Although the theme is an important for future pathophysiological elucidation of this disorders, it should be said that there were some critical concerns about the manuscript.

The most serious concern was that it was unclear what scientific approach this manuscript was based on. Although it appeared to cover a large number of studies, there was no description of what methods were used to conduct the review. Furthermore, a number of the authors' personal experienced cases that have not been published were included in what is presented as a review. It is unclear whether this manuscript was a review or a case report. The discussion mentioned that there is no gold standard for diagnosing functional neurological disorders, but clarification on the author's definition in this study would be appreciated. Did it include all conversion disorders in the DSM? Finally, the author seems to recommend throughout the manuscript that functional neurological disorders not be treated as psychiatric disorders based on the fact that biological biomarkers have been revealed, but this alone does not mean that they are not psychiatric disorders since many psychiatric disorders have also shown changes in brain function or structure.

Author Response

This manuscript, titled “Biomarkers and Rehabilitation for Functional Neurological Disorder,” deals with a review of functional neurological disorders. Although the theme is an important for future pathophysiological elucidation of this disorders, it should be said that there were some critical concerns about the manuscript.

The most serious concern was that it was unclear what scientific approach this manuscript was based on. Although it appeared to cover a large number of studies, there was no description of what methods were used to conduct the review. Furthermore, a number of the authors' personal experienced cases that have not been published were included in what is presented as a review. It is unclear whether this manuscript was a review or a case report. The discussion mentioned that there is no gold standard for diagnosing functional neurological disorders, but clarification on the author's definition in this study would be appreciated. Did it include all conversion disorders in the DSM? Finally, the author seems to recommend throughout the manuscript that functional neurological disorders not be treated as psychiatric disorders based on the fact that biological biomarkers have been revealed, but this alone does not mean that they are not psychiatric disorders since many psychiatric disorders have also shown changes in brain function or structure.

 The author thanks the reviewer for these important points, some of which are addressed in the comments to Reviewer 1. In response, the manuscript has now been revised to indicate the systematic review of FND and biomarkers, following a Boolean search of the PubMed citations.

The author’s including the illustrated cutaneous and orthopedic changes in the FND patients in his clinic are intended to complement the overall many overlaps between FND and various other neurological disorders. The cutaneous changes are so common in the author’s practice (previously reported as 17%, reference 134) that it was felt to be important to include and illustrate, because these changes had not been previously published, and thus to alert practitioners for this finding that may otherwise be overlooked or disregarded. Although research by the author on these changes is ongoing, the present invitation to the author to contribute to the Journal of Personal Medicine on the topic of FND, to provide to a general practitioner with an orientation toward the diverse multiple-organ changes in FND along with the extensive overlap of biomarker findings with conventional neurological disorders, the present author felt it important to include this material to bring to attention these important findings and their overlap with specific neurodegenerative disorders, particularly multiple system atrophy. Because these are cutaneous changes, illustration is needed.

This author had previously published his definition of FND in Mark VW, Functional neurological disorder: extending the diagnosis to other disorders, and proposing an alternate disease term—Attentionally-modifiable disorder. NeuroRehabilitation 2022;50:179-207. The present manuscript indicates in Section 2, lines 51-55, “The patients’ self-attention to their symptoms or emotional excitement can aggravate them, while distraction from them may reduce their severity [10]. FND symptoms can be provoked by direct medical examination and subside when the patient believes not to be observed or not undergoing formal evaluation [11-13].” To this passage, the author now adds the sentence, “These observations support diagnosing FND.” The statement is provided cautiously, because, as noted, there is as yet a gold standard for diagnosing FND, that is, a consensus set of criteria that would be endorsed by the majority of specialists who treat FND. At present, as shown by my paper with Brian Kirkwood (Consistency of inclusion criteria for functional movement disorder clinical research studies: a systematic review. NeuroRehabilitation 2022;50:169-78), there are three main, conflicting methods of diagnosis, which unfortunately continue to the present and hinder clinical research advances for this disorder.

Because the conversion disorder hypothesis is untestable (cf Carson A, et al. Psychologic theories in functional neurologic disorders. Handb Clin Neurol 2016;139:105-20), the present author does not apply the DSM method, or any other method, by which to diagnose “conversion disorder.” The term is unfortunately retained in the international classification system ICD-10 as the billing codes of F44.4, F44.5, F44.6, and F44.9.

The delineation between neurological and psychiatric disorder has been historically indistinct. In the 19th century, these medical areas were united. In the 20th century, they diverged. Thus, it is unclear why one should maintain that a classically psychiatric disorder such as schizophrenia today should not also be considered as a neurological disorder, given its biomarker findings. Consequently, it is unclear to the present author what is meant specifically by a “psychiatric disorder,” given that mood disturbances are common in conventional neurological disorders, including epilepsy (Fiest KM, et al. Depression in epilepsy: a systematic review and meta-analysis. Neurology 2013;80:590-599), dystonia (Lehn A, et al,  Psychiatric disorders in idiopathic-isolated focal dystonia. J Neurol 2014;261:668-74), Parkinson disease (Mele B, et al. Detecting anxiety in individuals with Parkinson disease. A systematic review. Neurology 2018;90:e39-e47), multiple system atrophy (Zhang LY, et al. Depression and anxiety in multiple system atrophy. Acta Neurol Scand 2018;137:33-7), and stroke (Campbell Burton CA, et al. Frequency of anxiety after stroke: a systematic review and meta-analysis of observational studies. Int J Stroke 2013;8:545-59).

The intent with the present manuscript is not to disregard and avoid treating mood disorder in persons with FND, but to support practitioners who may see persons with FND that it is not fundamentally a psychiatric disorder, any more than one can say for stroke etc. In response to this concern, the abstract is modified to read that “Functional neurological disorder, or FND, is widely misunderstood, particularly when considering recent research that indicating that the illness has a biological basis in addition to its psychiatric disorder association. Nonetheless, the long-held view that FND is a purely psychiatric disorder without biological basis, or even a contrived (malingered) illness, remains pervasive both in current medical care and general society.” This is important because at present psychiatric disorder is stigmatizing, as noted in the present manuscript. Consequently, persons who are diagnosed with FND are commonly belittled by their clinicians who do not specialize in this disorder, and not given appropriate medical care. Similarly, under section 6, Discussion, line 562, the statement is provided: “This evidentiary foundation allows practitioners to indicate to their patients that (1) the illness is not fundamentally a psychiatric disorder,…” This paragraph concludes (lines 567-569) to state, “Nonetheless, persons with FND who present with mood disorder should be treated for these symptoms by clinicians with expertise in either psychiatry or psychology.”

Round 2

Reviewer 2 Report

Comments and Suggestions for Authors

Many thanks to the authors for addressing many of the reviewer's comments. However, the following points require further improvements.
Since this paper was a systematic review, no content other than the results obtained by the author's specified literature search methodology should be presented. In other words, case reports that have not been published should be removed. It should also be described in the limitation section that the methodology of this systematic review was inadequate. Generally, literature extraction with only one researcher is inappropriate, and the opinion that the results are arbitrary is unavoidable.
While it is understandable that FND has a biological basis, whether it is classified as a neurological or psychiatric disorder seems to be a different debate because it is difficult to classify them, as the author points out. Therefore, throughout the manuscript, it would be better to describe the FND as a disease with a biological basis rather than a neurological disease.

Author Response

Reviewer 2, round 2 comments

Many thanks to the author for addressing many of the reviewer's comments. However, the following points require further improvements.
Since this paper was a systematic review, no content other than the results obtained by the author's specified literature search methodology should be presented. In other words, case reports that have not been published should be removed.

The author agrees that the review should be focused on the review itself and not include extraneous material. Therefore, unpublished observations from the author’s clinic are removed.

It should also be described in the limitation section that the methodology of this systematic review was inadequate. Generally, literature extraction with only one researcher is inappropriate, and the opinion that the results are arbitrary is unavoidable.

Limitations for the search method for this article are now described in lines 479-490.

While it is understandable that FND has a biological basis, whether it is classified as a neurological or psychiatric disorder seems to be a different debate because it is difficult to classify them, as the author points out. Therefore, throughout the manuscript, it would be better to describe the FND as a disease with a biological basis rather than a neurological disease.

The recommendations are appreciated. Accordingly, the manuscript is revised to draw attention to both the neurological as well as the other body systems changes that occur in FND.  Statements of the characteristics of FND that occur outside of the central nervous system are indicated in the Abstract (lines 21-23), Discussion (lines 498-505; lines 524-525).